# Nurse–physician collaboration, autonomy, and decision-making ability among intensive care unit nurses in Tehran, Iran: A cross-sectional study

Sogand Sarmadi[1], Neda Sanaie[2], Mahsa Boozari Pour[2,3], Akbar Zare-Kaseb[3] *

**1** Student Research Committee, Department of Medical Surgical Nursing, School of Nursing and Midwifery, Shahid Beheshti University of Medical Sciences, Tehran, Iran, **2** Department of Medical-Surgical Nursing, School of Nursing and Midwifery, Shahid Beheshti University of Medical Sciences, Tehran, Iran, **3** Clinical Research Development Center, Shahid Modarres Educational hospital, Shahid Beheshti University of Medical Sciences, Tehran, Iran

☯ All authors are co-corresponding authors, with equal responsibility.
* akbar.zarekaseb@gmail.com

## Abstract

### Background

Clinical decision-making by intensive care unit nurses is vital for ensuring patient safety. It is hypothesized that professional autonomy and interprofessional collaboration are determinants that influence the quality of decision-making; however, empirical evidence supporting this hypothesis within Iranian intensive care unit settings remains limited.

### Objective

To examine the correlations between perceived autonomy, nurse–physician collaboration, and clinical decision-making ability among intensive care unit nurses.

### Methods

This study employed a cross-sectional design, with 400 intensive care unit nurses from two university hospitals in Tehran completing validated Persian versions of the P-AITCS-II, the Dempster/DPBS, and the Lauri & Salanterä instruments to measure interprofessional collaboration, professional autonomy, and clinical decision-making, respectively. We followed STROBE reporting recommendations. Descriptive statistics characterized the sample and scale scores. Bivariate associations were assessed using correlation and group-comparison tests, and multiple linear regression was employed to estimate the independent associations of autonomy and collaboration with decision-making after adjusting for relevant covariates. Ethical approval and informed consent were obtained.

**Data availability statement:** The datasets generated and analyzed during the current study are not publicly available due to the restrictions imposed by the Modarres Clinical Research Development Center to ensure participant confidentiality and adherence to local laws. The raw dataset supporting the results of this study is securely stored. It can be made available to the Modarres Clinical Research Development Center (shahidmodarreshospital@gmail.com) upon reasonable request, provided that it is in accordance with the relevant ethical approvals and institutional regulations. The statistical analyses performed in SPSS are fully documented, allowing qualified researchers to verify and reproduce the results.

**Funding:** The author(s) received no specific funding for this work.

**Competing interests:** NO authors have competing interests.

## Results

The study sample comprised 400 participants, with a mean age of 33.19 years (SD = 5.67). In bivariate analyses, higher perceived autonomy and greater interprofessional collaboration were positively associated with better clinical decision-making ($p < 0.05$). In multivariable linear regression models that included covariates, perceived autonomy and interprofessional collaboration remained independent predictors of decision-making ability ($p < 0.05$). The entire model explained approximately 14.2% of the variance in decision-making, suggesting that additional determinants may influence nurses' decision-making processes. The total mean score of the participants was classified at an intuitive–analytic decision-making level, indicating a combined reliance on experience-based intuition and structured analytic processes.

## Conclusions

Among the intensive care unit nurses within this sample, higher perceived professional autonomy and enhanced interprofessional collaboration were independently correlated with improved self-reported clinical decision-making capabilities. Given the cross-sectional nature of the study, causality cannot be established. Longitudinal research employing objective performance metrics is necessary to examine causal relationships and to guide organizational strategies that foster both autonomy and collaborative practice.

## Introduction

The decision-making process involves cognitive functions such as problem identification, data collection and assessment, consequence evaluation, and optimal solution selection. This procedure employs recognized cognitive models like rational choice theory, where individuals decide by assessing the costs and benefits of different options [1]. Decision-making aptitude refers to an individual's ability to apply information, expertise, and problem-solving techniques to make informed choices in professional or clinical settings [2].

As a result of this process, two major cognitive modes are generally distinguished: analytical reasoning derived from systematic assessment and evidence, and intuitive reasoning derived from pattern recognition and experiential knowledge. According to the Intuitive-Analytic or Dual-Process theory of cognition, skilled nurses often integrate both modes simultaneously, resulting in intuitive and analytical decision-making. With this blended approach, it is possible to make rapid yet accurate judgments in dynamic, high-acuity environments, such as intensive care units (ICUs) [3,4].

Research demonstrates that nurses often report limited participation in critical decision-making, particularly in ICUs. This limited involvement primarily results from hierarchical organizational structures and physician-dominated decision-making processes. Nurses face challenges in maintaining professional autonomy due to these hierarchies, limited opportunities for influencing inter-professional collaboration, and

insufficient recognition of their expertise [5]. Targeted interventions, such as educational programs and supportive policies, can enhance nurses' professional competencies, thereby addressing these challenges. Educational programs and supportive policies represent such targeted interventions. This approach can enhance patient care outcomes [6,7].

Effective collaboration between nurses and physicians is crucial for delivering high-quality healthcare, particularly in intensive care units (ICUs), where clinical decisions have a direct impact on patient outcomes. Nurse-physician collaboration refers to a structured process of shared decision-making [8]. Interdisciplinary collaboration enables healthcare teams to address complex patient needs by fostering mutual respect, shared responsibilities, and effective communication. Nurses in the ICU frequently encounter high-stress situations and require autonomy in clinical judgment while maintaining effective participation within inter-professional teams. Collaboration between nurses and physicians facilitates shared decision-making and enhances the quality of patient care [9].

A study demonstrated that collaboration between ICU nurses and physicians was lower compared to the level of collaboration observed between palliative care physicians and nurses [10]. A study conducted in Iran reported moderate to high levels of inter-professional collaboration among healthcare professionals [11]. The implementation of structured communication strategies and inter-professional training enhances collaboration and improves team dynamics within healthcare settings [12].

Autonomy refers to the ability to manage individual professional responsibilities and exercise independent clinical judgment. It is an essential component of nursing practice. Autonomy supports critical thinking, adaptability, and the development of innovative approaches in patient care [13]. Nurses in critical care are essential for patient management, requiring advanced decision-making skills and executive competence. Recent studies have highlighted the influence of organizational culture, leadership styles, and interprofessional collaboration on nurses' autonomy. Supportive and participatory leadership in decision-making enhances nurses' autonomy and job satisfaction, which results in improved care outcomes [14].

Recent evidence from Iran and neighboring countries demonstrates significant links between interprofessional collaboration, professional autonomy, and nurses' decision-making processes. Cross-sectional and qualitative research have revealed variations in collaboration between nurses and physicians, as well as in nurses' perceptions of autonomy. Additionally, these studies have identified associations between autonomy, teamwork, moral distress, and clinical decision-making among critical care nurses. Based on these regional findings, it is evident that decision-making is not only influenced by individual cognitive strategies but also by organizational culture, leadership, and interprofessional relationships. This supports the need for examining nurse-physician collaboration and autonomy as predictors of decision-making in intensive care units [15–18].

It has been demonstrated that nurses' self-concept significantly influences their ability to make sound clinical decisions, underscoring the importance of considering personal and professional identity when studying decision-making. Similarly, research on the relationship between emotional intelligence and decision-making among NICU nurses has demonstrated that higher emotional intelligence is positively correlated with better decision-making abilities. It has also been reported that high levels of occupational stress were associated with reduced resilience among nurses in intensive care units during the COVID-19 pandemic, suggesting that psychological strain may impact cognitive flexibility and decision-making confidence during critical situations. Based on these findings, it appears that, in addition to organizational structures, individual characteristics such as self-concept, emotional intelligence, and resilience interact directly with professional autonomy and interprofessional collaboration to influence decision-making ability at the personal level [19–21].

Several previous studies have demonstrated that structured interactions and high-quality communication between physicians and nurses not only enhance the quantity and speed of clinical information exchange but also create conditions that enable nurses to act independently and with greater authority by strengthening psychological empowerment and legitimizing their clinical roles. It has been demonstrated in reviews and empirical studies in ICU settings that nurses' perceived autonomy is associated with improved clinical decision-making and patient safety. Strong team relationships

and organizational support can strengthen or weaken this relationship. Clinical reasoning suggests that nurses can shift between intuitive and analytical decision-making strategies adaptively, based on their access to team information and the ability to act autonomously, as demonstrated by cognitive models such as the Recognition-Primed Decision model, which has been validated in high-risk clinical settings. This theoretical mechanism is based upon the proposition that collaboration reduces uncertainty by increasing informational and social support, thereby enhancing psychological empowerment and perceived autonomy, thereby improving clinical decision-making capacity [22–26].

Despite previous research on nurses' autonomy, inter-professional collaboration, and decision-making, substantial gaps persist. Much of the study concentrates on isolated elements, lacking a comprehensive examination of their interconnections. Additionally, there is a paucity of research on how collaboration affects the development of critical thinking and decision-making skills, which have frequently been studied separately [27]. Empirical evidence regarding the influence of leadership styles and resource accessibility on inter-professional collaboration, autonomy, and decision-making capabilities is lacking [28]. Considering the unique organizational hierarchies, cultural factors, and resource constraints in Iranian hospitals, understanding these relationships is essential for designing targeted interventions to empower nurses and improve patient care outcomes.

The purpose of this study was to evaluate nurse-physician collaboration, perceived autonomy, and decision-making ability among critical care nurses in Iranian ICUs. Based on the literature review and identified research gaps, the study aimed to answer the following research questions:

What is the level of nurse-physician collaboration among critical care nurses in Iranian ICUs?

What is the level of perceived autonomy among these nurses?

What is the decision-making ability of ICU nurses, and how do collaboration, autonomy and sociodemographic factors influence it?

## Methods

### Study design and setting

This cross-sectional analytical study conformed to the guidelines stipulated by the Strengthening the Reporting of Observational Studies in Epidemiology (STROBE) (S1 Appendix). It was conducted in 2025 at two hospitals in Tehran affiliated with Shahid Beheshti University of Medical Sciences, namely Imam Hossein and Loghman Hakim. These hospitals serve as tertiary care teaching institutions providing a broad range of inpatient and outpatient services, including critical care units. Sampling was conducted in eight ICUs across the two hospitals: Imam Hossein (five ICUs) and Loghman Hakim (three ICUs).

### Participants

The study involved ICU nurses who met the inclusion criteria. Participants were recruited using convenience sampling from August 18, 2025, to September 8, 2025.

### Inclusion criteria

The inclusion criteria comprised full-time employment as an ICU nurse at Imam Hossein and Loghman Hakim hospitals, possession of at least a Bachelor of Science degree in Nursing, a minimum of one year of professional experience in ICUs, and voluntary participation, as evidenced by providing informed consent.

### Exclusion criteria

The exclusion criteria encompassed incomplete or missing questionnaire responses; duplicate registrations or submissions, including the transmission of identical content from multiple IP addresses; and low-quality responses, such as straight-lining, excessively rapid completion, or evidence of internal inconsistencies. These criteria were implemented

to ensure the integrity and quality of the collected data. Straight-lining responses indicate inattentive or non-thoughtful answering, while duplicate registrations could bias the results by overrepresenting certain participants.

## Sample size

The aim of establishing the relationship between nurse-physician collaboration and nurse decision-making independence guided the determination of the sample size. Within this framework, the data originated from a similar study, which revealed a correlation of 0.242 between nurse-physician collaboration and nurses' independent decision-making capabilities [29]. Employing the sample size formula for correlation, with 95% confidence and 90% power, the requisite sample size, based on the minimum correlation of 0.242, was determined to be 164 individuals. Accounting for a 10% dropout rate, the calculation yielded a result of 180 participants. Based on an a priori calculation, 180 participants would be needed to detect a medium effect size for our primary outcome with conventional Type I and Type II error rates. The recruitment rate during the data collection period exceeded the initial target for several practical reasons: a higher-than-anticipated response rate, active engagement and endorsement by unit head nurses, an extended recruitment window, and participation from additional ICUs that agreed to participate. All completed questionnaires from eligible participants were retained, resulting in a final sample of 400 nurses. We retained the larger sample to increase the precision of estimates, improve the stability and generalizability of results, and improve statistical precision and power for multivariable analyses.

## Data collection

Following the arrangements with the university and receipt of the study's ethical code, sampling was conducted in the two hospitals. The researchers developed the questionnaires online via the (www.porsline.ir) website. Following initial contact, the researchers explained the study's purpose and methodology to the department's head nurses and subsequently obtained their approval to distribute the questionnaires. Researchers then evaluated the online completion process for the questionnaire. Before distributing the questionnaires, the head nurses explained the study's aim, the questionnaire's structure, and the completion guidelines to the ICU nurses. Subsequently, the questionnaire link was distributed to the target populations through the "Bale" and "WhatsApp" applications, and nurses were encouraged to complete it. Completing the questionnaire was mandatory to prevent data loss, and respondents could not proceed without fully addressing all items. It was only possible to submit the questionnaire response from a specific Internet Protocol (IP) address once. To minimize potential biases associated with mandatory online completion, participants were assured of anonymity and confidentiality, and instructions emphasized the importance of honest and accurate responses.

## Questionnaires

The research employed four questionnaires: 1) Demographics, 2) Interprofessional Team Collaboration Scale II (P-AITCS-II), 3) Dempster Professional Behaviour Scale (DPBS), and 4) Lauri and Salanterä Clinical Decision-Making Questionnaire (LSCD-MQ).

## Demographic Questionnaire

This survey collected data on age, gender, marital status, employment status, work experience, monthly shift averages, work shifts, financial status, and level of education.

## Interprofessional Team Collaboration Scale II (P-AITCS-II)

The initial version of this questionnaire was developed by Orchard et al. from Canada and includes 47 questions [30]. In 2025, Norouzinia et al. conducted a psychometric validation of the Persian translation of the Inter-professional Team

Collaboration Assessment Index (Inter-professional Team Collaboration Questionnaire, Version II (P-AITCS-II)) for health-care providers. The Persian version of this questionnaire comprises 23 items and is divided into three sub-dimensions. This survey utilizes a 5-point Likert scale, with values ranging from 1 to 5, anchored by "strongly disagree" and "strongly agree". Exploratory factor analysis of the Persian version of this questionnaire yielded three extracted factors: participation, collaboration, and coordination. Moreover, the confirmatory factor analysis results indicated an acceptable model fit: CMIN/DF = 2.41, CFI = 0.917, RMSEA = 0.079. The Cronbach's alpha coefficients for each dimension were computed as 0.88, 0.90, and 0.93. In our study, the overall reliability of the questionnaire was acceptable, with a Cronbach's alpha of 0.89. The reliability coefficients for the sub-dimensions of participation, collaboration, and coordination were 0.87, 0.88, and 0.88, respectively.

### Dempster Professional Behavior Scale (DPBS) (Autonomy)

The Dempster Professional Behavior Scale (DPBS), was initially developed by Dempster et al. (31) and utilized in the research conducted by Agha Mohammadi et al. [31]. This scale is one of the most widely used instruments for evaluating autonomy in nursing. It has been tested across diverse clinical contexts. Over nearly three decades, the DPBS has consistently demonstrated validity and reliability in measuring professional autonomy among nurses [32]. In addition, secondary analyses have supported its ability to capture essential dimensions of autonomy, including independent decision-making, accountability, and professional judgment [33]. More recently, cross-cultural validation studies have confirmed its applicability in international nursing contexts, strengthening its relevance for contemporary research [34]. Therefore, the DPBS provides a comprehensive and psychometrically sound instrument for evaluating the autonomy of critical care nurses in this study. This instrument consists of thirty items, categorized into four subscales: readiness, empowerment, fulfillment, and valuation. Each item is rated on a Likert scale ranging from 1 to 5, where 1 indicates complete falsehood and 5 indicates complete truth. The preliminary version of this instrument demonstrated acceptable internal consistency, with a Cronbach's alpha of 0.95, and showed moderate to strong correlations, indicating empirical one-dimensionality. Higher total scores on the instrument indicate greater autonomy. The study by Agha Mohammadi reported a content validity index (S-CVI) of 0.95 and a Cronbach's alpha coefficient of 0.83 for this questionnaire. In our study, the overall reliability of the DPBS was acceptable, with a Cronbach's alpha of 0.85. The reliability coefficients for the subscales of readiness, empowerment, fulfillment, and valuation were 0.90, 0.89, 0.90, and 0.86, respectively.

### Lauri and Salanterä Clinical Decision-Making Questionnaire (LSCD-MQ)

Lauri and Salanterä initially developed the Lauri and Salanterä Clinical Decision-Making Questionnaire (LSCD-MQ) to evaluate nurses' decision-making capabilities [35,36]. This instrument`s Persian version incorporates 24 items related to clinical decisions and is assessed using a 5-point Likert scale, ranging from "always = 5" to "never = 1". Therefore, the total score for this instrument ranges from 24 to 120. Scores below 68 reflect analytical decision-making (Level One), scores between 68 and 78 indicate intuitive-analytical decision-making (Level Two), and scores exceeding 78 represent intuitive decision-making (Level Three). This instrument was designed according to four phases of the decision-making process: (a) data collection, (b) information review and problem identification, (c) planning and implementation, and (d) follow-up and evaluation. In Iran, several studies have evaluated the reliability and validity of the Persian version of the Lauri Clinical Decision-Making Questionnaire. Ghodsi Astan et al. reported a Cronbach's alpha of 0.81, confirming its reliability [37,38]. As part of Javadi's study (1389/2010), the questionnaire was submitted to a panel of nursing faculty members and English-language experts, who confirmed its content validity. An internal consistency analysis was conducted on a sample of nurses to determine the instrument's reliability, yielding a Cronbach's alpha of 0.78 [39]. In a separate study carried out in 2014, the Lauri instrument was translated into Persian by the researcher in collaboration with an English-language faculty member, followed by a back-translation into English. After reviewing and addressing translation discrepancies, the pre-final version was submitted to a panel of ten nursing faculty members for evaluation of content validity. The

 

internal consistency of the final Persian questionnaire was assessed using Cronbach's alpha, which was reported to be α = 0.85 [40]. The overall reliability of the LSCD-MQ was acceptable, with a Cronbach's alpha of 0.79. The reliability coefficients for the four phases—data collection, information review and problem identification, planning and implementation, and follow-up and evaluation—were 0.82, 0.84, 0.85, and 0.83, respectively.

## Ethical consideration

Before the commencement of sampling, an ethical code was obtained from the Modarres Clinical Research Development Center, Shahid Beheshti University of Medical Sciences (IR.SBMU.RETECH.REC.1404.360). Sampling was started with the university's approval and coordination with the nursing management at the specified hospitals. The study's participation was voluntary, and participants remained anonymous. Informed consent was obtained through a link that covered the study's aims and methods, potential risks and benefits, privacy protocols, anonymity guarantees, and voluntary participation. The ethical guidelines of the Declaration of Helsinki (2013 revision) governed this investigation [41,42]. Using an online platform that does not collect personally identifiable information, all responses were anonymized and confidential. Participants were assigned unique study IDs, and the resulting datasets were stored on a secure, password-protected server accessible only to the research team. As a result of these procedures, participants' identities remained confidential and data handling was carried out in accordance with ethical principles.

## Data analysis

**Descriptive statistics.** Descriptive statistics were applied to the quantitative data analysis. This included the mean and standard deviation for data exhibiting a symmetrical distribution. Percentages and numerical values were used to present the categorical data.

**Normality checks.** The normality of the data was assessed using both graphical methods (histogram and Q-Q plot) and statistical tests (Shapiro–Wilk test) (S2 Appendix).

**Correlation analyses.** The Pearson correlation coefficient was reported for the main study variables. The relationships between the investigated main variables and quantitative demographic variables were also analyzed using correlation coefficients. In contrast, categorical demographic variables were assessed via two-way independent t-tests and multi-way analysis of variance (ANOVA).

**Multiple regression models.** General linear modeling was implemented to control for confounders when investigating the relationship between variables. Demographic variables were initially collected broadly, but only those variables that showed statistically significant associations in preliminary analyses (p < 0.05) were included as covariates in the multivariable analyses. This approach allowed us to control for potential confounding effects while focusing on variables that were meaningfully related to decision-making ability. This study used Stata version 16 software. The tests were conducted using a 5% significance level and were bilateral.

## Results

### Sample characteristics

The study sample consisted of 400 participants. Most were female (272 individuals, 68.0%) and married (265 individuals, 66.25%). Regarding work schedules, most participants worked rotating shifts (210 individuals, 52.50%), while 104 individuals (26.00%) worked extended day shifts and 86 individuals (21.50%) worked night shifts. Financial status was predominantly reported as "good" by 241 respondents (60.25%), with 83 respondents (20.75%) indicating a weak economic status and 76 respondents (19.00%) reporting a moderate financial status. The majority of participants held a bachelor's degree (303 individuals, 75.75%), followed by 91 individuals (22.75%) with a master's degree, and six individuals (1.50%) with a PhD. Additionally, a substantial proportion of respondents (317 individuals, 79.25%) reported holding supplementary job in addition to their primary occupation.

For continuous variables, the mean age was 33.19 years (SD = 5.67). The mean ICU experience was 10.14 years (SD = 6.19), and the average number of monthly shifts was 18.34 (SD = 3.94). These sociodemographic and professional characteristics are summarized in Table 1.

### Decision-making ability, nurse–physician collaboration, and autonomy

**Descriptive statistics.** The overall Nurse–Physician Collaboration score averaged $M = 69.01$ ($SD = 17.70$), with its subscales showing similar central tendencies: Partnership, M = 23.99 (SD = 8.25), Cooperation, M = 24.02 (SD = 8.36), and Coordination, $M = 20.99$ ($SD = 7.55$) (See Table 2).

Autonomy recorded the highest mean among the principal constructs (M = 90.28, SD = 18.71). Its constituent scores were as follows: Readiness, M = 32.91 (SD = 11.21); Empowerment, M = 21.06 (SD = 7.63); Fulfillment, M = 27.27 (SD = 9.43); and Valuation, M = 9.05 (SD = 3.76). Decision-Making exhibited a mean of 71.99 (SD = 14.05), with its subdomains detailed as follows: Data collection, M = 18.01 (SD = 6.22); Information review and problem identification, M = 18.04 (SD = 6.38); Planning and implementation, M = 17.95 (SD = 6.52); and Follow-up and evaluation, M = 17.99 (SD = 6.29). Although the number of participants classified at the first level (Analytical level) of decision-making was slightly higher than others, the mean score in the studied population falls within the Intuitive-Analytical level (Level 2). No significant differences were observed among participants in terms of decision-making levels.

**Correlations.** Collaboration demonstrated highly significant and positive associations with its subscales: Cooperation (r = 0.780, p < .001), Partnership (r = 0.710, p < 0.001), and Coordination (r = 0.704, p < 0.001). Additionally, collaboration was positively correlated with Autonomy (r = 0.531, p < 0.001), Empowerment (r = 0.385, p < 0.001), Fulfilment (r = 0.322, p < 0.001), Valuation (r = 0.244, p < .001), and Decision-Making (r = 0.272, p < 0.001), suggesting that enhancing collaboration may support nurses' Autonomy and Decision-Making Ability.

**Table 1. Socio-demographic characteristics of the sample. ICU: intensive care unit; PhD: Doctor of Philosophy; SD: Standard Deviation.**

| Variable (Categorical) | | Frequency | Percentage |
|---|---|---|---|
| **Sex** | **Female** | 272 | 68.0 |
| | **Male** | 128 | 32.0 |
| **Marital Status** | **Married** | 265 | 66.25 |
| | **Single** | 135 | 33.75 |
| **Work Shift** | **Long Day** | 104 | 26.00 |
| | **Night** | 86 | 21.50 |
| | **Rotating** | 210 | 52.50 |
| **Financial Status** | **Weak** | 83 | 20.75 |
| | **Moderate** | 76 | 19.00 |
| | **Good** | 241 | 60.25 |
| **Last Education Level** | **Bachelor`s** | 303 | 75.75 |
| | **Master`s** | 91 | 22.75 |
| | **PhD** | 6 | 1.50 |
| **Other Jobs** | **No** | 83 | 20.75 |
| | **Yes** | 317 | 79.25 |
| **Variable (Continuous)** | | **Mean** | **SD** |
| **Age (Years)** | | 33.19 | 5.67 |
| **ICU Experience (Years)** | | 10.14 | 6.19 |
| **Monthly Shift (Numbers in Month)** | | 18.34 | 3.94 |

Table 2. Descriptive Statistics for Major Study Variables.

| Variable | Observation | Mean | SD |
|---|---|---|---|
| **Nurse–Physician Collaboration** | **400** | **69.005** | **17.698** |
| Partnership | 400 | 23.997 | 8.245 |
| Cooperation | 400 | 24.023 | 8.363 |
| Coordination | 400 | 20.985 | 7.549 |
| **Autonomy** | **400** | **90.282** | **18.705** |
| Readiness | 400 | 32.905 | 11.212 |
| Empowerment | 400 | 21.063 | 7.629 |
| Fulfillment | 400 | 27.265 | 9.428 |
| Valuation | 400 | 9.05 | 3.755 |
| **Decision-Making** | **400** | **71.987** | **14.051** |
| Data collection | 400 | 18.012 | 6.218 |
| Information review and problem identification | 400 | 18.04 | 6.383 |
| Planning and implementation | 400 | 17.95 | 6.519 |
| Follow-up and evaluation | 400 | 17.985 | 6.289 |
| **Decision-Making Levels** | **Observation** | **N** | **%** |
| Analytical (Level 1) | 400 | 146 | 36.50 |
| Intuitive-Analytical (Level 2) | 400 | 126 | 31.50 |
| Intuitive (Level 3) | 400 | 128 | 32.00 |

Autonomy exhibited a robust association with its component, Readiness (r = 0.700, p < 0.001), and demonstrated significant positive correlations with Fulfilment (r = 0.605, p < 0.001) and Empowerment (r = 0.546, p < 0.001). Furthermore, Autonomy was moderately correlated with Decision-Making (r = 0.348, p < 0.001).

Decision-Making exhibited strong positive correlations with its four subdomains: Planning and implementation (r = 0.606, p < 0.001), Data collection (r = 0.553, p < 0.001), Follow-up and evaluation (r = 0.529, p < .001), and Information review and problem identification (r = 0.522, p < .001).

In contrast, demographic and work-experience variables (Age, ICU experience, and Monthly shifts) demonstrated no meaningful or statistically significant relationships with the principal studied variables. For instance, Age correlations were negligible and non-significant, as were ICU experience and Monthly shifts. The detailed results of the correlations are presented in Table 3.

**Comparison regarding sample characteristics.** Overall, no statistically significant differences were observed in collaboration, autonomy, or decision-making based on gender or marital status. Similarly, analyses considering academic level and financial status did not reveal significant differences across the three principal variables.

The shift type was the sole characteristic that demonstrated a statistically significant difference: an ANOVA revealed a significant effect of shift type on decision-making (p = 0.046), indicating that nurses working rotating shifts may exhibit higher decision-making performance. In contrast, shift type was not significantly associated with autonomy (p = 0.199) or collaboration (p = 0.895). Furthermore, having another job exhibited a non-significant trend regarding decision-making (p = 0.064), and was unrelated to autonomy (p = 0.720) and collaboration (p = 0.551) (See Table 4).

**Predictors of decision-making ability.** The diagram for the conceptual framework of the utilized model is presented in Fig 1. The complete model was statistically significant, $F_{(4, 395)} = 16.33$, p < 0.001, and explained approximately 14.2% of the variance in decision-making scores ($R^2 = 0.1419$, adjusted $R^2 = 0.1332$).

Among the predictors, autonomy emerged as the most influential and reliable determinant: each incremental increase of one unit in autonomy correlated with a 0.204-point enhancement in decision-making (unstandardized B = 0.204, standardized β = 0.272, standard error = 0.0416), t = 4.91, p < .001. Additionally, collaboration was identified as a significant

**Table 3. Correlation Matrix of Nurse-Physician Collaboration, Autonomy, and Decision-Making Ability.**

| Variables | (1) | (2) | (3) | (4) | (5) | (6) | (7) | (8) | (9) | (10) | (11) | (12) | (13) | (14) | (15) | (16) | (17) |
|---|---|---|---|---|---|---|---|---|---|---|---|---|---|---|---|---|---|
| (1) Nurse–Physician Collaboration | 1.000 | | | | | | | | | | | | | | | | |
| (2) Partnership | **0.710** (0.000) | 1.000 | | | | | | | | | | | | | | | |
| (3) Cooperation | **0.780** (0.000) | **0.320** (0.000) | 1.000 | | | | | | | | | | | | | | |
| (4) Coordination | **0.704** (0.000) | **0.218** (0.000) | **0.372** (0.000) | 1.000 | | | | | | | | | | | | | |
| (5) Autonomy | **0.531** (0.000) | **0.331** (0.000) | **0.445** (0.000) | **0.391** (0.000) | 1.000 | | | | | | | | | | | | |
| (6) Readiness | **0.272** (0.000) | **0.213** (0.000) | **0.206** (0.000) | **0.177** (0.000) | **0.700** (0.000) | 1.000 | | | | | | | | | | | |
| (7) Empowerment | **0.385** (0.000) | **0.240** (0.000) | **0.341** (0.000) | **0.264** (0.000) | **0.546** (0.000) | **0.166** (0.001) | 1.000 | | | | | | | | | | |
| (8) Fulfilment | **0.322** (0.000) | **0.157** (0.002) | **0.287** (0.000) | **0.266** (0.000) | **0.605** (0.000) | 0.087 (0.081) | 0.051 (0.307) | 1.000 | | | | | | | | | |
| (9) Valuation | **0.244** (0.000) | **0.131** (0.009) | **0.190** (0.000) | **0.218** (0.000) | **0.263** (0.000) | -0.056 (0.262) | 0.065 (0.194) | **0.138** (0.006) | 1.000 | | | | | | | | |
| (10) Decision-Making | **0.272** (0.000) | **0.135** (0.007) | **0.254** (0.000) | **0.210** (0.000) | **0.348** (0.000) | **0.184** (0.000) | **0.299** (0.000) | **0.170** (0.001) | **0.151** (0.002) | 1.000 | | | | | | | |
| (11) Data collection | **0.135** (0.007) | 0.037 (0.461) | **0.134** (0.002) | **0.127** (0.011) | **0.140** (0.005) | 0.064 (0.205) | **0.120** (0.016) | 0.062 (0.219) | **0.107** (0.033) | **0.553** (0.000) | 1.000 | | | | | | |
| (12) Information review and problem identification | 0.085 (0.089) | 0.047 (0.347) | **0.128** (0.007) | 0.007 (0.888) | **0.164** (0.001) | 0.072 (0.284) | **0.126** (0.016) | **0.129** (0.010) | 0.021 (0.681) | **0.522** (0.000) | 0.043 (0.392) | 1.000 | | | | | |
| (13) Planning and implementation | **0.223** (0.000) | **0.187** (0.000) | **0.151** (0.002) | **0.151** (0.002) | **0.294** (0.000) | **0.213** (0.000) | **0.191** (0.000) | **0.133** (0.008) | **0.105** (0.036) | **0.606** (0.000) | **0.138** (0.006) | 0.084 (0.095) | 1.000 | | | | |
| (14) Follow-up and evaluation | **0.157** (0.002) | 0.023 (0.648) | **0.148** (0.003) | **0.179** (0.000) | **0.170** (0.001) | 0.054 (0.324) | **0.223** (0.000) | 0.051 (0.309) | **0.103** (0.040) | **0.529** (0.000) | 0.061 (0.227) | 0.022 (0.655) | 0.096 (0.055) | 1.000 | | | |
| (15) Age | -0.033 (0.509) | | | | -0.049 (0.332) | | | | | -0.030 (0.550) | | | | | 1.000 | | |
| (16) ICU Experience | -0.053 (0.292) | | | | -0.054 (0.278) | | | | | -0.042 (0.407) | | | | | | 1.000 | |
| (17) Monthly Shift | -0.013 (0.802) | | | | -0.049 (0.324) | | | | | -0.060 (0.228) | | | | | | | 1.000 |

**Table 4. Comparison of nurse-physician collaboration, autonomy, and decision-making ability by sample characteristics (N = 400). ᵃIndependent t-tests ᵇAnalysis of variance (ANOVA).**

| Characteristics | N | Collaboration | P value | Autonomy | P value | Decision-making | P value |
|---|---|---|---|---|---|---|---|
| **Gender** | | | | | | | |
| Male | 128 | 67.63 (18.77) | **0.285**[a] | 88.81 (19.02) | **0.282** [a] | 71.55 (12.76) | **0.673** [a] |
| Female | 272 | 59.65 (17.17) | | 90.97 (18.55) | | 72.19 (14.63) | |
| **Marital status** | | | | | | | |
| Single | 135 | 70.55 (18.23) | **0.212** [a] | 91.70 (17.96) | **0.277** [a] | 72.51(14.58) | **0.590** [a] |
| Married | 265 | 68.21 (17.40) | | 89.55 (19.06) | | 71.72 (13.80) | |
| **Academic Level** | 400 | | **0.556** [b] | | **0.171** [b] | | **0.341** [b] |
| **Financial Status** | 400 | | **0.264** [b] | | **0.719** [b] | | **0.678** [b] |
| **Shift Type** | 400 | | **0.895** [b] | | **0.199** [b] | | **0.0458*[b]** |
| **Other Job** | | | | | | | |
| No | 317 | 68.73 (17.35) | **0.551** [a] | 90.45 (18.51) | **0.720** [a] | 72.65 (14.15) | **0.064** [a] |
| Yes | 83 | 70.03 (19.01) | | 89.62 (19.53) | | 69.44 (13.42) | |

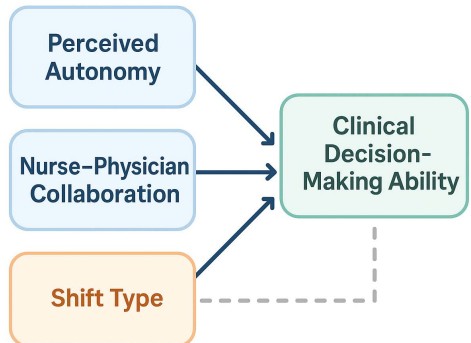

**Fig 1. The diagram for the conceptual framework of the utilized model.**

positive predictor: a one-unit increase in collaboration was associated with a 0.102-point increase in decision-making (B = 0.102, β = 0.128, standard error = 0.043), t = 2.32, p = .021.

Regarding shift type (reference = long day shift), the rotating shift was correlated with higher decision-making scores (B = 3.29, β = 0.117, SE = 1.58), t = 2.09, p = .037. The night shift indicated a positive but statistically non-significant association (B = 2.97, β = 0.087, SE = 1.91), t = 1.55, p = .121. The results of the multiple regression analysis are presented in Table 5.

## Discussion

This cross-sectional study examined nurse–physician collaboration, perceived autonomy, and clinical decision-making ability among critical care nurses. It examined whether collaboration and autonomy serve as predictors of decision-making after controlling for shift type. The study also found that nurses' mean score of clinical decision-making was at the level of intuitive-analytical decision-making (Level Two). The perceived autonomy among ICU nurses was above average, as was the level of perceived collaboration. Correlation and regression analyses revealed that autonomy and collaboration were positively correlated with decision-making, and both variables independently predicted decision-making within a multivariable model. The comprehensive model accounted for 14.2% of the variance in decision-making. Rotating shifts

**Table 5. Multiple Regression Predicting Decision-Making Ability Among Critical Care Nurses. Number of observations 400, F (4, 395) = 16.33, Prob > F = 0.0000, R-squared 0.1419, Adj R-squared 0.1332.**

| Variable | | Coefficient | Standardized Beta | Std. error | t | P>|t| |
|---|---|---|---|---|---|---|
| Autonomy | | .204 | .272 | .0416 | 4.91 | <0.001 |
| Nurse–Physician Collaboration | | .102 | .128 | .043 | 2.32 | 0.021 |
| Work Shift | Night | 2.97 | .087 | 1.91 | 1.55 | 0.121 |
| | Rotating | 3.29 | .117 | 1.58 | 2.09 | 0.037 |
| Constant | | 44.18 | | 3.50 | 12.61 | <0.001 |

(versus long day shifts) were associated with a modestly higher decision-making score. The effect of rotating shifts on decision-making observed in this study can be theoretically explained by several mechanisms supported in recent literature. First, rotating shifts exposes nurses to a wider variety of clinical presentations and diverse team members, which can broaden experiential knowledge and pattern recognition — crucial for effective intuitive-analytical decision-making. Second, rotating schedules are strongly linked to sleep disruption, fatigue, and increased perceived stress, which impair attention, working memory, and executive functions—all of which are essential for analytic processing and safe clinical judgment. Third, rotating shifts may also lead to variable exposure to leadership styles, procedural norms, and decision-support tools across shifts, reinforcing adaptive flexibility but potentially introducing inconsistency in cues and expectations. Thus, rotating shifts likely amplify both opportunities (experiential breadth, exposure) and risks (fatigue, inconsistent context), and their net effect on decision-making depends on how these opposing forces balance in a given unit [43–45].

Other demographic and workload indicators, including age, ICU experience, and monthly shifts, were not significantly related to the primary study variables. The lack of significant associations between demographic variables and the primary constructs may be due to the relative homogeneity of the sample or other sampling characteristics.

Practical nurse–physician communication is widely recognized as a crucial element of safe and high-quality critical care. Deficiencies in teamwork and information exchange are frequently associated with adverse events and compromised patient outcomes. Interprofessional collaboration holds particular significance in the ICU, where rapid fluctuations in patient condition necessitate prompt and precise handovers, as well as transparent role negotiation. Systematic reviews and syntheses on ICU settings highlight that cohesive interprofessional teams improve care coordination and facilitate effective clinical decision-making [46]. The study conducted by Degu et al. in 2023 revealed that nurse-physician collaboration was below anticipated levels; consequently, a substantial number of participants experienced ineffective collaborations [47]. This does not align with the findings of our study, as the level of collaboration observed was favorable. This discrepancy may be attributed to our research, specifically designed and conducted among ICU nurses. In contrast, the referenced study included nurses from various hospital wards. This variation might indicate higher levels of collaboration within the ICU. In this context, Parizad et al. (2021) conducted a study on nurses in ICU settings that aligns with our results. Their research assessed nurses' job stress and its relationship with professional autonomy and nurse–physician collaboration in the ICU. The findings revealed that nurse–physician collaboration was at desirable levels, which may confirm the presence of higher interdisciplinary collaboration in wards where the patient acuity necessitates increased collaboration [29]. Boev et al. (2022) qualitatively documented a consensus among nurses and physicians regarding the importance of effective interprofessional communication for interprofessional collaboration, explicitly connecting collaborative practice to improved patient outcomes. Their participants delineated distinct yet complementary priorities: nurses emphasized the necessity of mutual respect, while physicians highlighted the significance of robust professional relationships. Both groups identified multidisciplinary rounds as the primary forum for operationalizing collaboration. These qualitative insights support our quantitative findings, indicating that the interpersonal and organizational facets of collaboration

observed in practice are aligned with measurable improvements in nurses' clinical decision-making [48]. Evidence from various contexts underscores persistent barriers to fully actualizing nurse–physician collaboration. A cross-sectional study conducted in hospitals in southern Lebanon in 2020 aimed to explore attitudes toward the physician–nurse relationship and revealed that nearly a quarter of physicians disagreed with the notion that nurses should be regarded as collaborators and colleagues in patient care. The study further demonstrated that nurses reported higher collaboration scores than physicians, with female participants achieving higher scores than their male counterparts. Nevertheless, the overall collaboration scores within this sample were markedly lower than those documented in comparable international research. These findings suggest that hierarchical perceptions, gender disparities, and systemic deficiencies persistently impact interprofessional dynamics within the region. The authors recommended enhancing collaboration through interprofessional education and training at both undergraduate and postgraduate levels. Collectively, these insights underscore the need for cultural and structural initiatives to promote mutual respect and effective communication, thereby reinforcing our finding that increased collaboration is positively correlated with improved decision-making among critical care nurses [49].

Abate et al. (2022) conducted a cross-sectional study to evaluate the clinical decision-making approaches used by hospital nurses. The findings indicated that nearly half of the participants employed intuitive decision-making strategies, while the remaining half utilized analytical strategies. These results align with those of our own study, which revealed that the nurses in the study population demonstrated a comparable distribution of decision-making levels. This consistency suggests that nurses generally integrate experience-based pattern recognition (intuition) with deliberate data collection and evaluation (analysis) when making clinical judgments—an integrated approach well-suited to the fast-paced yet information-rich environment of the ICU. Practically, the prevalence of an intuitive-analytical approach underscores the importance of interventions that both enhance experiential learning—through mentoring, high-quality clinical exposure, and simulation—and strengthen analytical skills via structured assessment tools and decision aids, rather than favoring one approach exclusively. Furthermore, given that our regression analysis demonstrates a positive association between greater autonomy and collaboration with higher decision-making scores, creating workplace conditions that promote independent judgment and effective information exchange may facilitate nurses in deploying their intuitive and analytical skills more reliably and safely [50].

Our findings are also consistent with evidence from other contexts. For example, Abu Arra et al. conducted a study in Palestinian hospitals to identify factors influencing nurses' clinical decision-making in emergency departments. They reported that the mean score for clinical decision-making was slightly above average, indicating that nurses demonstrated a moderate to high level of decision-making competence. This pattern aligns with the present study, in which ICU nurses similarly scored above the midpoint on decision-making measures. These findings suggest that although nurses in various healthcare systems attain at least an average level of decision-making proficiency, there remains potential for further enhancement of clinical judgment through targeted educational and organizational interventions [51]. This research additionally determined that the level of nursing education and working hours serve as predictors of clinical decision-making among nurses employed in emergency departments. Regarding work shifts and hours, the study's findings align with those of the survey conducted in the ICU. However, our investigation revealed no significant correlation between clinical decision-making and educational attainment. [51]. A likely explanation in the Iranian context is that the number of nurses with advanced degrees (Master's or PhD) working in clinical settings is relatively small, as most pursue academic or teaching positions rather than direct patient care. Therefore, the influence of formal education on bedside decision-making may be limited in this setting.

According to Benner's Novice-to-Expert framework, nurses increasingly integrate pattern recognition (intuition) with analytical reasoning as they progress through stages of clinical competence. This suggests that the prevalence of intuitive-analytical decision-making modes in our sample is consistent with Benner's framework. According to Benner, expert nurses can recognize familiar clinical patterns rapidly while still applying analytical processes when the situation is novel or uncertain. Due to the high level of technology and information density in the ICU, a purely intuitive approach

may not provide the necessary rigor for complex clinical problems, whereas an analytical approach may be too slow or impractical under time pressure. Therefore, the mixed mode (intuitive + analytical) is not only plausible but also likely to be optimal in dealing with dynamic and ambiguous ICU contexts. It is evident from recent analyses of ICU decision-making that nurses in critical care settings must alternate between rapid pattern recognition and deliberate reasoning, depending on the task demands and the complexity of the patient [4]. Furthermore, new conceptualizations, such as the Person-Centered Nursing Model and the Cognitive Continuum Theory, suggest that decision-making processes are based on a spectrum of emotions and change according to cues, workload, and experience. When Benner's theory is integrated with these cognitive models, the interpretation of our finding becomes stronger: the intuitive-analytic blend reflects adaptive expertise appropriate to the technological, fast-paced environment associated with ICUs [52].

In the present study, nurses exhibited relatively high levels of autonomy, which also emerged as the strongest predictor of decision-making ability. This result builds upon the findings of numerous prior studies, which generally reported autonomy at moderate levels. For example, Alshaikh et al. [53] found that the majority of nurses had moderate autonomy, Afifa et al. [28] reported similar results, with half of the participants falling within the moderate range, Saeed et al. [54] showed that most of the participants exhibited moderate professional autonomy, and Gharaaghaji et al. [55] also described autonomy as moderate in their cohort. By contrast, Kurt et al. [56] documented high autonomy among nurses, a finding that is more consistent with our results. These comparisons suggest that although many nursing populations operate within a moderate spectrum of autonomy, critical care nurses in our study may encounter greater autonomy. Several contextual factors may contribute to our sample's comparatively high autonomy score. As a result of the distinctive clinical requirements of intensive care, including rapid patient deterioration, frequent bedside problem-solving, and reliance on experienced clinical judgment, nurses can exercise greater independence in decision-making, as evidenced by reviews of ICU nursing autonomy [22]. Furthermore, concept analyses and qualitative studies indicate that ICU culture (shared expectations about role breadth and local protocols that empower bedside nurses) may contribute to nurses' perception of their autonomy in the ICU [57]. It is also likely that the organizational context, including hospital type and interprofessional practice, contributes. Studies have indicated that nurse participation and perceived autonomy differ between institutions, with clinical rounds and decision forums that are actively supported by nursing staff showing higher autonomy and participation scores than those that are not [58]. Finally, historical patterns of role expectations and local nurse-physician hierarchies continue to influence the situation. In areas with lower hierarchical barriers and strong collaborative norms, nurses report greater freedom to make clinical decisions. However, in regions with systemic or hierarchical barriers, autonomy is usually restricted [59]. Our findings of above-average autonomy among ICU nurses in this sample are plausible explanations of these contextual factors taken together.

Pursio et al.'s 2021 study aimed to synthesize existing knowledge on professional autonomy in nursing, identifying themes such as shared leadership, professional competencies, interprofessional and intraprofessional collaboration, and maintaining a healthy work environment [60]. The quantitative outcomes of our research underscore the importance of interprofessional collaboration as a key factor influencing professional autonomy. Additionally, the findings of the studies conducted by Parizad et al. and Mohamed et al. further corroborate this association [29,61].

## Strengths

This study utilizes a sufficiently large and appropriate sample size (N = 400), ensuring robust statistical power to identify associations and facilitate multivariable modeling. The instruments employed are the Persian adaptations with documented validity and reliability, thereby reinforcing the precision of construct measurement. The analytical methodology integrated correlation analyses, group comparisons, and multiple regression techniques to control for potential confounding variables and to independently assess the effects of professional autonomy and interprofessional collaboration on clinical decision-making. By focusing on ICU nurses and utilizing multidimensional subscales for core constructs, the study

enables a more detailed understanding of underlying mechanisms. Lastly, adherence to ethical procedures and reporting standards (STROBE) enhances methodological transparency and the quality of reporting.

## Limitations

The cross-sectional design precludes causal inference. Observed associations between autonomy/collaboration and decision-making may be bidirectional or confounded by unmeasured factors. A longitudinal design should be employed in future research to explore causal relationships and better disentangle the directionality of associations.

Convenience sampling and online questionnaire distribution in two university hospitals in Tehran introduce selection bias and limit the generalizability of the findings to other regions or health systems. Conducting multi-center, larger, and regionally diverse studies to enhance external validity and generalizability across different healthcare settings is recommended.

This study relied on self-administered questionnaires to measure nurses' decision-making, autonomy, and collaboration, which may introduce self-report bias. Participants might have overestimated or underestimated their actual behaviors or perceptions due to social desirability or recall effects. Although anonymity and confidentiality were assured to reduce this risk, the influence of self-report bias cannot be entirely ruled out; objective performance measures or direct observational data would complement subjective reports.

Using online data collection only and mandating complete responses on the survey platform may introduce response biases such as social desirability. Although the Persian versions of the instruments used in this study have demonstrated acceptable validity and reliability in Iranian samples, it is undeniable that cultural factors, language nuances, and local ICU practices may influence the respondents' responses. It is essential to consider these potential limitations when interpreting the results, and future studies should investigate the cross-cultural applicability of these tools in greater detail.

It is worth noting that only a small fraction of the decision-making variance was explained by the included predictors. Significant remaining variability is likely to be accounted for by unmeasured individual, situational, and organizational factors (as well as measurement limitations of self-report scales). A significant portion of decision-making variance remains unexplainable, despite the importance of autonomy and collaboration. For future research to be effective, additional cognitive, psychological, situational, and organizational variables should be incorporated. Future studies utilizing longitudinal designs, mixed qualitative and quantitative approaches, advanced analytical methods, and objective indicators will be more effective in disentangling sources of variance.

## Conclusion

Among ICU nurses, higher perceived professional autonomy and greater interprofessional collaboration were independently associated with improved clinical decision-making, collectively explaining a significant, though not comprehensive, 14% of the variance in decision-making ability. Participants demonstrated a relatively equal distribution in terms of the level of decision-making. The overall mean score for the targeted population falls within the intuitive-analytic approach, indicating that nurses depend on a combination of experience-based intuition and structured analytical processes in their clinical judgment. These findings have important clinical and systemic implications, suggesting that enhancing professional autonomy and fostering interprofessional collaboration can contribute not only to safer and more effective patient care but also to broader organizational improvements in ICU practice. Targeted organizational, educational, and policy interventions based on these findings may facilitate evidence-informed, collaborative decision-making, ultimately supporting patient safety and quality of care.

### Implications for clinical practice

It is recommended that pragmatic, context-sensitive organizational and educational measures be implemented in the ICUs of Tehran hospitals to enhance clinical decision-making. In terms of organizational implementation, hospitals should pilot shared governance structures and regular, time-protected interprofessional case-review forums in one or two ICUs before

extending the program. Nurses and physicians should be appointed as champions, and multidisciplinary case reviews should be conducted weekly for 30–60 minutes with explicit role assignments for nurses. Simple decision-pathway templates should also be introduced to clarify when nurses are authorized to initiate specific interventions. Monitoring performance should be accomplished using feasible indicators (e.g., the proportion of nurse-initiated actions documented at the bedside, the completeness of handover checklists, and selected patient safety indicators). Potential obstacles, such as high workload, limited staffing, resource constraints, and resistance to organizational change, may affect the implementation of these measures and should be carefully considered in planning.

## Recommendations for future research

It is recommended that future research build upon the findings and limitations of the present study by using a more comprehensive and contextually relevant approach. To better capture the influence of organizational factors, such as managerial support, workload, and institutional culture, it is recommended that these factors be incorporated into predictive models of clinical decision-making. To enhance measurement validity, it is also essential to use objective and multi-source data, including direct observations, chart audits, and patient safety indicators, in combination with self-report instruments.

It would be beneficial to conduct replication studies with larger, multi-center, or multi-regional samples in order to improve generalizability across different hospital settings and populations. Furthermore, longitudinal designs are required for examining causal relationships between professional autonomy, interprofessional collaboration, and clinical decision-making. Ultimately, the use of mixed methods can contribute to the elucidation of contextual mechanisms, the identification of mediators and moderators, and the exploration of cultural or institutional influences on decision-making processes.

## Implications for education

Educational training should explicitly integrate intuitive and analytical reasoning. This can be achieved by implementing short, repetitive simulation sessions (including low-cost scenario drills and structured debriefing), case-based reflection rounds, and mentorship pairings that incorporate experiential learning and explicit instruction on clinical reasoning heuristics and data-driven decision aids. It is possible to stage these interventions as small pilots, evaluate them using the same autonomy, collaboration, and decision-making criteria used in this study, and adapt them iteratively to meet staffing and resource constraints. Together, these steps aim to create an environment where professional autonomy and effective collaboration can coexist, contributing to safer and more reliable decisions in the ICUs. Practical challenges, including staff time constraints and availability of training resources, should be anticipated when implementing these educational interventions.

## Recommendations for policy and nursing management

Iranian hospitals should implement policies and procedures that explicitly acknowledge and safeguard nurses' decision-making authority, clearly defining their scope of practice and responsibilities in accordance with national regulations and hospital protocols. It is essential to systematically integrate metrics for professional autonomy and interprofessional collaboration into quality assurance and accountability systems, ensuring effective monitoring and continuous improvement.

Hospitals are encouraged to invest in continuous professional development, sustain appropriate staffing ratios, and allocate resources that promote prompt communication among care teams. Practical strategies may include appointing clinical champions, implementing structured interprofessional case review meetings, and introducing decision-support tools to aid nurse-led interventions.

The implementation process should take into account potential institutional, cultural, and logistical challenges prevalent in Iranian ICUs. These include hierarchical decision-making practices, limited staffing resources, high patient workload, and varying degrees of managerial support. Conducting pilot interventions within selected units, complemented by training

programs and feedback mechanisms, can effectively address these barriers. Such efforts are instrumental in fostering a progressively collaborative, evidence-based culture that reinforces clinical governance, promotes patient safety, and enhances the quality of care.

## Supporting information

**S1 Appendix. STROBE Statement.**
(DOCX)

**S2 Appendix. Graphical representations (histogram and Q-Q plot) for checking the normality assumption and results of normality tests.**
(DOCX)

## Acknowledgments

The researchers would like to take this opportunity to express their sincere gratitude to the Dean of the Clinical Research Development Center of Shahid Modarres Educational Hospital, Shahid Beheshti University of Medical Sciences, for their support of this study, without which this research would not have been possible.

## Author contributions

**Conceptualization:** Sogand Sarmadi, Neda Sanaie, Akbar Zare-Kaseb.

**Data curation:** Sogand Sarmadi, Mahsa Boozari Pour, Akbar Zare-Kaseb.

**Formal analysis:** Sogand Sarmadi, Neda Sanaie, Akbar Zare-Kaseb.

**Funding acquisition:** Sogand Sarmadi, Neda Sanaie, Akbar Zare-Kaseb.

**Investigation:** Mahsa Boozari Pour.

**Methodology:** Sogand Sarmadi, Akbar Zare-Kaseb.

**Project administration:** Mahsa Boozari Pour.

**Resources:** Sogand Sarmadi, Akbar Zare-Kaseb.

**Software:** Mahsa Boozari Pour.

**Supervision:** Sogand Sarmadi, Neda Sanaie, Akbar Zare-Kaseb.

**Validation:** Sogand Sarmadi, Mahsa Boozari Pour, Akbar Zare-Kaseb.

**Visualization:** Sogand Sarmadi, Neda Sanaie, Mahsa Boozari Pour.

**Writing – original draft:** Sogand Sarmadi, Neda Sanaie, Akbar Zare-Kaseb.

**Writing – review & editing:** Sogand Sarmadi, Neda Sanaie, Mahsa Boozari Pour, Akbar Zare-Kaseb.

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
