## [Decision Letter · Decision Letter 0]

13 Oct 2025

Dear Dr. Zare-Kaseb,

Thank you for submitting your manuscript to PLOS ONE. After careful consideration, we feel that it has merit but does not fully meet PLOS ONE’s publication criteria as it currently stands. Therefore, we invite you to submit a revised version of the manuscript that addresses the points raised during the review process.

We look forward to receiving your revised manuscript.

Kind regards,

Federica Canzan

Academic Editor

PLOS ONE

Journal Requirements:

2. Please provide additional details regarding participant consent. In the ethics statement in the Methods and online submission information, please ensure that you have specified (1) whether consent was informed and (2) what type you obtained (for instance, written or verbal, and if verbal, how it was documented and witnessed).

4. In the online submission form you indicate that your data is not available for proprietary reasons and have provided a contact point for accessing this data. Please note that your current contact point is a co-author on this manuscript. According to our Data Policy, the contact point must not be an author on the manuscript and must be an institutional contact, ideally not an individual. Please revise your data statement to a non-author institutional point of contact, such as a data access or ethics committee, and send this to us via return email. Please also include contact information for the third party organization, and please include the full citation of where the data can be found.

6. Please include your tables as part of your main manuscript and remove the individual files. Please note that supplementary tables (should remain/ be uploaded) as separate "supporting information" files.

Reviewers' comments:

Reviewer's Responses to Questions

**Comments to the Author**

1. Is the manuscript technically sound, and do the data support the conclusions?

Reviewer #1: Yes

Reviewer #2: Yes

Reviewer #3: Partly

2. Has the statistical analysis been performed appropriately and rigorously?

Reviewer #1: Yes

Reviewer #2: Yes

Reviewer #3: Yes

3. Have the authors made all data underlying the findings in their manuscript fully available?

Reviewer #1: Yes

Reviewer #2: Yes

Reviewer #3: Yes

4. Is the manuscript presented in an intelligible fashion and written in standard English?

Reviewer #1: Yes

Reviewer #2: Yes

Reviewer #3: Yes

Reviewer #1: Introduction

Incorporating recent regional studies will enhance relevance and underscore the multidimensionality of decision-making. This insertion would fit at the end of the Introduction section, just before stating the study aim.

Recent evidence highlights the importance of individual psychosocial factors in shaping clinical decision-making among nurses. For example, Aboalrob et al. (2025) demonstrated that nurses’ self-concept significantly influences their ability to make sound clinical decisions, underscoring the need to consider personal and professional identity in decision-making research. Similarly, Ayed (2025) investigated the relationship between emotional intelligence and decision-making among NICU nurses, reporting that higher emotional intelligence was positively associated with better decision-making competence. These findings suggest that beyond organizational structures, individual-level attributes such as self-concept and emotional intelligence interact with professional autonomy and interprofessional collaboration to shape decision-making ability.

Methods

The convenience sampling from two hospitals may limit external validity. Please clarify how many ICUs/units were covered in each hospital and whether participants were evenly distributed.

Instruments

More detail is needed on the cultural adaptation process of the Lauri & Salanterä tool in Persian ICU contexts.

Results and Discussion

Please emphasize why autonomy scores were higher in this sample compared to previous studies, and discuss how local ICU culture, nurse–physician hierarchy, or hospital type may have contributed.

Avoid overemphasis on non-significant demographic findings.

Implications

Implications for practice and policy should be more practical and context-specific. For instance, could structure interprofessional rounds, shared governance, or simulation-based training realistically be implemented in Tehran ICUs?

Educational recommendations should emphasize decision-making training that integrates intuitive and analytical reasoning, as noted in recent regional literature.

Limitations

The limitations section should be expanded to include Self-report bias

Aboalrob W, Ayed A, Malak MZ, Aqtam I. Understanding the influence of self-concept on clinical decision-making among nurses: A cross-sectional study. Plos one. 2025 Aug 25;20(8):e0330905.

Ayed A. The relationship between the emotional intelligence and clinical decision-making among nurses in neonatal intensive care units. SAGE Open Nursing. 2025 Feb;11:23779608251321352.

Reviewer #2: I am grateful for the opportunity to review the paper "Nurse-physician Collaboration, Autonomy, and Decision-making Ability among Critical Care Nurses: A Cross-sectional Study." This study examines a highly relevant aspect of critical care nursing with a high sample size and standardized tools to determine significant correlations. Overall, the paper is well presented and clear to read. However, certain methodological, analytical, and interpretative issues must be resolved to enhance the validity of the findings and the overall impact of the paper.

Reviewer #3: TITLE AND ABSTRACT

1. The title is clear and reflects the scope, but the phrase “A Cross-sectional Study” could be expanded with a more specific methodological explanation or mention of the country (e.g., “in Iran,” etc.).

INTRODUCTION

1. The introduction presents the topic of “decision-making in critical care nurses” with very broad transitions; key concepts (e.g., “intuitive–analytic decision-making”) should be defined first, followed by their place in the literature.

2. Although the literature review is comprehensive, some sections are written using too many first-person pronouns or vague expressions; they should be presented using more objective, straightforward sentences.

3. The conceptual framework should be clarified with a model or diagram; a logical flow should be added. The connection between decision-making, autonomy, and collaboration should be shown more clearly.

4. In the literature review, a brief definition should be provided for each key concept (collaboration, autonomy, decision-making), and emphasis should be placed on why these are important for ICU nurses.

5. The research gap and problem definition presented are not expressed in a sufficiently direct and original manner; for example, comparisons with the Iranian context or previous studies are made arbitrarily, and specific reasons and shortcomings should have been emphasized more sharply.

6. A specific paragraph stating “the purpose of the study” should be added directly at the end of the introduction; this is not clearly distinguished in the current text. The sentence “Therefore, this study was conducted” should be directly linked to the purpose, and if high-impact results are expected, these should be clearly addressed in the text.

7. The research question or hypotheses section should be formulated clearly and concisely at the end of the introduction.

8. The reference list is insufficient, especially in terms of new and local studies; it should be further supported by current literature (last 2 years) and should be based on local data as well as classical studies.

9. Comparisons with international literature are particularly lacking in the citation system; for example, findings from other countries should be added in addition to “previous Iranian studies.”

METHODS

1. The study design is described as “cross-sectional analytical,” but despite being multicenter, it does not explicitly discuss the limitation of generalizability.

2. Participant selection was performed using convenience sampling; this issue has not been sufficiently addressed in terms of bias risk, external validity limitations, and sample representativeness. The exclusion criteria (“straight-lining,” “duplicate registration”) lack methodological justification; the extent to which they ensure sample integrity is debatable. Sample selection and exclusion criteria should be discussed in more detail in terms of cause-and-effect relationships.

3. Only online completion was used in the data collection process, and the possible impact of this method on social desirability and response bias was not mentioned.

4. The data collection procedure (“mandatory online completion”) carries risks of accuracy and bias due to the sample; the possibility of social desirability bias should be discussed. The fact that participants submitted only once from the same IP address does not in itself ensure reliability. Strategies for addressing the risks of bias in online data collection should be clearly stated.

5. The validations of the scales used are well explained, but validity-reliability issues that may arise when applied in different cultures and local samples have not been fully addressed.

6. The demographic survey is broad in scope, but the role of the variables obtained in the analysis plan is not sufficiently explained.

7. It is positive that a power analysis was performed, but the sample size does not contribute to generalizability in practice, as it is limited to two hospitals.

8. Normal distribution checks and verification methods are explained, but alternative tests to be applied in case of deviation are not specified.

9. In multiple regression, the variables included in the model are clearly listed, but the rationale for not including additional variables is lacking, as the model's variance explanation rate is low (approximately 14%).

10. The SPSS version is specified, but the analysis processes are not detailed; for example, details such as which variables were used in which analysis and which control variables were applied for correlation are unclear. The statistical analysis section should provide detailed subheadings for all methods and control variables used. Additional control steps regarding the samples should be written to ensure the adequacy of the data set and its resistance to manipulation.

11. The ethical approval and consent process is clearly explained, but the steps taken for data anonymization and confidentiality are not described in technical detail.

12. Restrictions on data sharing are specified, but there is no additional information regarding access to raw data and the verifiability of the analyses.

FINDINGS

1. The findings were initially presented using descriptive statistics (socio-demographic characteristics, means, standard deviations), but in their current form, they provide excessive detail; the main text includes unnecessary repetition in the explanations of the tables.

2. Under each subheading (e.g., correlation analysis, group comparisons, regression), the coherence between text and tables is weak; the matching of findings to tables and the explanatory notes should be improved.

3. The flow of findings is disjointed; it is not clearly emphasized which analysis responds to which hypothesis.

4. Table titles and variable descriptions are sometimes missing or confusing; in particular, sub-dimension names and abbreviations should be clear to an international audience.

5. References to tables and figures are not provided appropriately between findings; clear references to tables should be made within the text (e.g., “See Table 2”).

6. Descriptive statistics only provide the mean and SD; in some cases, percentages and frequency distributions should also be included.

7. The explanatory power of the multiple regression analysis is low (14.2%), but the text lacks a reference to the reason for this or to the limitations section.

8. The practical meaning of both descriptive and inferential statistics should be contextualized with short summary sentences; for example, the relationship between statistically significant differences and clinical/practical meaning should have been discussed.

9. The tables and statistical analyses are adequate, but in some places the descriptive statistics take up too much space, and the findings section is unnecessarily long. The findings should be condensed.

10. The regression model explains only 14.2% of the variance; this indicates that other important variables (organizational culture, managerial support, workload, etc.) are missing and should be clearly discussed.

11. Some variable names in the table presentations are mistranslated or confusing (e.g., “Collaboration” and its sub-dimensions). Internationally accepted titles and abbreviation definitions should be added for tables and graphs.

12. Some tables and appendices are not fully integrated into the main text.

13. Some findings (e.g., the relationship between educational status and decision-making) are said to be inconsistent with previous studies, but the reasons are not sufficiently discussed.

14. The results regarding differences between different groups (e.g., shift type, educational status) are left without justification, i.e., the literature review related to the findings is lacking.

15. The finding of no relationship between demographic variables and main variables is presented without discussion; however, the homogeneity of the sample or any sampling bias is not addressed.

16. If there are findings/statistics regarding the limitations of “self-report” data, they are not emphasized; possible biases such as social desirability bias could be briefly mentioned here before leaving them for discussion.

17. Findings/analysis processes should be titled according to the hypothesis or research questions (e.g., “H1: Relationship between ... and ...”, “H2: Difference between ... groups”, etc.).

18. In addition to significance values (p), additional statistical indicators such as effect size (Cohen’s d, OR, Beta) and confidence intervals should be provided if available.

DISCUSSION

1. The main findings are repeated but not discussed in a focused manner according to the research question or hypotheses; that is, a systematic analysis of the findings according to the predetermined objectives/hypotheses is lacking.

2. Although literature comparisons were adequately performed, the country-specific (Iran) system, cultural, and institutional differences that could lead to similar/different findings should have been discussed more robustly.

3. Contradictory results in the literature are approached only at the level of “we agree/disagree”; the possible contradictions between the findings and contextual reasons are not explored in depth.

4. The limitations section is presented positively; however, methodological constraints such as the “self-report” method, sample representativeness, and social desirability bias should be supported with more concrete data and examples.

5. The ability to generalize the findings should be discussed in detail; the sample structure and distribution should be elaborated beyond the statement that it is “limited to two hospitals.”

6. The recommendations provided for future research are too general and do not directly address the findings and limitations of the study; more specific, actionable, and gap-filling recommendations are needed (e.g., incorporating organizational factors into future models, supporting them with observational data).

7. Repetition with large-scale, multi-center, and/or multi-regional samples should be recommended.

8. Limitations are correctly stated, but social desirability, selection bias, and the use of self-reported data only should be emphasized more strongly. Leadership, organizational support, and organizational environment variables should have been modeled.

9. Clinical, educational, and management/policy-based recommendations are included, but the potential obstacles that may be encountered in implementing each recommendation or situations that may challenge existing practices at the country level have not been discussed.

10. The discussion lacks a “paradigm shift in clinical practice” or scientific justification for policy recommendations; the recommendations remain largely at a universal level.

11. The findings' “practice and policy” recommendations should be more concrete and detailed for institutions.

12. The implementation recommendations remain at a general level; more specific and practically applicable recommendations (e.g., “making interdisciplinary training mandatory”) could be presented.

13. The clinical and societal significance of the findings, their contribution to current practices, or how they could play a role in systemic improvement could be presented in greater detail.

14. The distinction between the conclusion paragraph and the discussion is unclear; a strong closing section summarizing the importance of the main findings, their limitations, and the recommendations should be added.

15. The reasons why the findings of the study may be only “partially” generalizable should be discussed more clearly; in particular, limitations in the sample, data collection method, and measurement tools may cast doubt on the reliability of the results.

16. Analysis and literature comparison should be structured according to hypotheses/objectives.

17. The limitations section should be diversified with concrete examples (e.g., for decision-making performance, not only self-reporting but also objective data suggestions).

18. Practical application and policy recommendations should include discussion of the specific context (country, institutional structure, culture), feasibility, and potential difficulties that may be encountered.

19. In the conclusion section, the impact of the study findings on the literature, nursing education, and management should be summarized and emphasized.

**Do you want your identity to be public for this peer review?** For information about this choice, including consent withdrawal, please see our Privacy Policy

Reviewer #1: No

Reviewer #2: No

Reviewer #3: No

---

## [Author Response · Author response to Decision Letter 1]

14 Oct 2025

The full response is provided in attachments.

---

## [Decision Letter · Decision Letter 1]

27 Oct 2025

Nurse–Physician Collaboration, Autonomy, and Decision-Making Ability among Intensive Care Unit Nurses in Tehran, Iran: A Cross-Sectional Study

PONE-D-25-48900R1

Dear Dr. Zare-Kaseb,

We’re pleased to inform you that your manuscript has been judged scientifically suitable for publication and will be formally accepted for publication once it meets all outstanding technical requirements.

Kind regards,

Federica Canzan

Academic Editor

PLOS ONE

Additional Editor Comments (optional):

Reviewers' comments:

Reviewer's Responses to Questions

**Comments to the Author**

Reviewer #1: All comments have been addressed

Reviewer #2: All comments have been addressed

2. Is the manuscript technically sound, and do the data support the conclusions?

Reviewer #1: Yes

Reviewer #2: Yes

3. Has the statistical analysis been performed appropriately and rigorously?

Reviewer #1: Yes

Reviewer #2: Yes

4. Have the authors made all data underlying the findings in their manuscript fully available?

Reviewer #1: Yes

Reviewer #2: Yes

5. Is the manuscript presented in an intelligible fashion and written in standard English?

Reviewer #1: Yes

Reviewer #2: (No Response)

Reviewer #1: the authors have adequately addressed my comments raised in a previous round of review and I feel that this manuscript is now acceptable for publication

Reviewer #2: The authors have done an excellent job addressing the reviewers' specific points, the manuscript is in a very strong position and suitable for publication

**Do you want your identity to be public for this peer review?** For information about this choice, including consent withdrawal, please see our Privacy Policy

Reviewer #1: No

Reviewer #2: No

---

## [Editor Report · Acceptance letter]

PONE-D-25-48900R1

PLOS ONE

Dear Dr. Zare-Kaseb,

I'm pleased to inform you that your manuscript has been deemed suitable for publication in PLOS ONE. Congratulations! Your manuscript is now being handed over to our production team.

Kind regards,

on behalf of

Professor Federica Canzan

Academic Editor

PLOS ONE